# Preservation of Litchi Fruit with Nanosilver Composite Particles (Ag-NP) and Resistance against *Peronophythora litchi*

**DOI:** 10.3390/foods11192934

**Published:** 2022-09-20

**Authors:** Xiaojie Lin, Yongsheng Lin, Zhengping Liao, Xianqian Niu, Yingxiang Wu, Dandan Shao, Bingrong Shen, Tingting Shen, Fang Wang, Hongyang Ding, Binji Ye, Yongyu Li

**Affiliations:** 1Institute of Natural Products of Horticultural Plants, Fujian Agriculture and Forestry University, Fuzhou 350002, China; 2Fujian Institute of Tropical Crops, Zhangzhou 363001, China; 3Qingyuan Agricultural Science and Technology Extension Service Center, Qingyuan 511500, China; 4Agricultural and Rural Bureau of Zhaoan County, Zhaoan 363500, China

**Keywords:** litchi, post harvest, nanosized silver, *Peronophythora litchi*, antifungal potential

## Abstract

Litchi (*Litchi chinensis* Sonn.) is susceptible to infection by *Peronophythora litchi* post storage, which rapidly decreases the sensory and nutritional quality of the fruit. In this study, the effects of nanosilver (Ag-NP) solution treatment on the shelf life of litchi fruit and the inhibition of *P. litchi* were examined, and the underlying mechanisms were discussed. For investigations, we used one variety of litchi (‘Feizixiao’), dipping it in different concentrations of Ag-NP solution after harvesting. Meanwhile, we treated *P. litchi* with different concentrations of Ag-NP solution. According to the data analysis, litchi treated with 400 μg/mL Ag-NPs and stored at 4 °C had the highest health rate and the lowest browning index among all the samples. In the same trend, treatment with 400 μg/mL Ag-NPs produced the best results for anthocyanin content, total soluble solids content, and titratable acidity content. Additionally, according to the results of the inhibition test, 800 μg/mL Ag-NP solution had a 94.97% inhibition rate against *P. litchi*. Within 2–10 h following exposure to 400 μg/mL Ag-NP solution, the contents of superoxide dismutase, peroxidase, and catalase in *P. litchi* gradually increased and peaked, followed by a gradual decline. At this time, the integrity of the cell membrane of *P. litchi* could be broken by Ag-NP solution, and the sporangia showed deformed germ tubes and abnormal shapes. Taken together, these results suggested that Ag-NP treatment inhibited respiration and *P. litchi* activity, which might attenuate litchi pericarp browning and prolong the shelf life of litchi. Accordingly, Ag-NPs could be used as an effective antistaling agent in litchi fruit and as an ecofriendly fungicide for the post-harvest control of litchi downy blight. This study provides new insights into the application of Ag-NP as an antistaling agent for fruit storage and as an ecofriendly fungicide.

## 1. Introduction

Litchi (*Litchi chinensis* Sonn.) is a common subtropical fruit that is loved by consumers [1]. Litchi post-harvest physiological and metabolic activities are exuberant and belong to the nonbreathing peak fruit [2]. Thus, litchi are very susceptible to biochemical and structural changes starting from the moment they are harvested, resulting in the rapid browning of the pericarp two-three days after harvest [3]. These post-harvest changes can be principally accelerated by the respiration of litchi and the action of microorganisms, mostly *P. litchi*. In a previous report, litchi losses caused by litchi downy blight and decay accounted for more than 20% of the total output every year [4], which resulted in significant challenges to the post-harvest preservation and transportation of litchi.

Previously, the fresh-keeping technologies of litchi were mainly low-temperature preservation [5], controlled-atmosphere preservation [6], and chemical preservation [7]. Among them, low-temperature fresh-keeping technology has been widely used. However, in the air-flow freezing system, air as a coolant cannot quickly cool the fruit, and it easily damages the structure and tissue of the fruit and affects the quality of litchi [8]. Controlled-atmosphere preservation can effectively preserve the post-harvest quality of litchi; in contrast, the construction cost of controlled-atmosphere storage is high, and gas parameters such as O_2_ and CO_2_ need to be strictly regulated, so it is not easy to popularize [9]. Although soaking litchi with chemical agents has a good fresh-keeping effect, the chemicals infiltrate into the pulp with the extension of soaking time, and there are some food safety risks [10].

Silver nanoparticles (Ag-NPs) are a safe and nontoxic new material [11] with a high specific surface area, multiporous structure, and strong catalytic and bactericidal properties [12]. According to recent research, Ag-NPs have been considered promising materials for improving the safety of fruits and vegetables and extending their shelf life by protecting them from external environmental factors. Additionally, Ag-NPs have strong inhibition and killing effects on dozens of pathogenic microorganisms [13] and do not produce drug resistance. Subsequently, Ag-NP materials are widely used in the preservation of fruit and vegetables and have been developed into antimicrobial agents [14]. The chitosan film containing Ag-NPs could prolong the shelf life of fresh-cut melon by more than thirteen days [15]. Ag-NP coatings could effectively prolong the shelf life of grape [16], jujube, and pistachio [17]. Ag-NP antibacterial packaging film could maintain the moisture of cabbage and tomato, prolonging the storage time [18]. Additionally, Ag-NPs could inhibit the activity of *Alternaria solani* and reduce the occurrence of early blight disease in tomato [19].

In our previous work, we succeeded in the green synthesis of Ag-NPs using extracts gained from *Phyllanthus emblica* fruit [20] and researched the effect of Ag-NP preservation of litchi fruit [21]. However, limited information is available regarding the inhibition of litchi downy blight and preservation via post-harvest Ag-NP treatment and its related mechanisms in litchi fruit. In this study, we examine whether Ag-NPs influence the post-harvest shelf life of litchi fruits and the activity of *P. litchi*. Additionally, the study provides preliminary insights into the underlying mechanisms this process. The purpose is to establish a more effective fruit storage system that aims to reduce the deterioration of post-harvest quality, reduce the loss rate, and prolong the shelf life.

## 2. Materials and Methods

### 2.1. Materials and Treatments

#### 2.1.1. Preparation of Ag-NP Solution

According to the method by Liao et al. [20], fresh and disease-free fruit of *Phyllanthus emblica* were washed, dried, enucleated, and freeze-dried, and the powder was obtained after sifting through forty meshes. The supernatant obtained from the powder extracted using hot water for five h was dehydrated step by step with ethanol and dried repeatedly with ether to obtain the crude polysaccharide.

A volume of 50 mL of 0.05 mol/L *Phyllanthus emblica* crude polysaccharide aqueous solution, 10 mL of 1 g/L NaCl, and 10 mL of silver nitrate solution were mixed [20]. The solution pH was adjusted to the desired value using ammonia solution. The mixture was stirred for 4 h at room temperature, and ultraviolet radiation at a wavelength of 365 nm was used to assist in the generation of Ag-NPs. After centrifuging the Ag-NP solution for 10 min at 10,000× *g* rpm, Ag-NPs were allowed to settle to the bottom of the conical tube. The supernatant phase was removed, and Ag-NPs were washed with 10 mL of distilled water for three times. After washing, the residue was freeze-dried and placed in a refrigerator at −40 °C for later use.

Ag-NPs were prepared in solutions of 0.5, 1, 2, 4, 8, and 16 mg/mL with distilled water.

#### 2.1.2. Ag-NP Treatment of Litchi

‘Feizixiao’ litchi (*Litchi chinensis* Sonn.) at commercial maturity (about 90 days after anthesis) was purchased from Yonghui Supermarket in Fuzhou, Fujian Province, China. We selected fruits of uniform size, and the surface of the fruits did not show obvious signs of disease or mechanical damage. The fruits were placed in a low-temperature foam box and immediately transported to Institute of Natural Products of Horticultural Plants of Fujian Agriculture and Forestry University (Fuzhou, Fujian). The stalks of the fruits were cut off, washed with distilled water, and dried with absorbent paper. The fruits were divided into six groups with 1000 litchis in each group and immersed into Ag-NP solutions at different concentrations (100, 200, 400, and 800 μg/mL) and control solution for 5 min each. We used deionized water and *Phyllanthus emblica* crude polysaccharide aqueous as controls. Then, all treated fruits were allowed to air-dry at room temperature. For each group, one thousand litchis were packed in one hundred polyvinyl chloride boxes. After packing, each litchi group was equally divided into two parts. One part was stored at room temperature for five days, and we collected samples every day for experimentation. The other parts were stored at 4 °C for twenty-five days, and samples were taken every five days for the experiment. Statistics were needed for each litchi box in each group when determining the health rate and browning index of litchi. During each experiment, two litchi were randomly selected from each of fifty boxes of litchi, and a total of one hundred litchi were used for the determination of corresponding indices.

#### 2.1.3. Ag-NP Treatment of P. litchi

*P. litchi* was isolated and preserved at Institute of Natural Products of Horticultural Plants of Fujian Agriculture and Forestry University (Fuzhou, Fujian); it was inoculated in V8 solid medium and grown at 30 °C for 7 d.

A total of 120 mL of V8 culture medium was poured into 6 culture plates; a volume of 1 mL was taken out from the medium, and 1 mL of Ag-NP solution at different concentrations (0.5, 1, 2, 4, 8, and 16 mg/mL) were added. Then, the concentrations of the final Ag-NP solutions of V8 culture media was 25, 50, 100, 200, 400, and 800 μg/mL. After the culture media were solidified, *P. litchi* was inoculated in different V8 media with a 5 mm hole punch. The same V8 solid medium without Ag-NP solution was used as a blank control. All V8 media were inverted-cultured for 5 d at 25 °C.

### 2.2. Estimation of the Health Rate, Browning Index, Anthocyanin Content, TSS, and TA of Litchi

The health rate, browning index, anthocyanin content, TSS, and TA of treated fruits were estimated following each storage period. According to the method by Yang et al. [22], the health rate of litchi after treatment was observed and counted based on the standard that the diameter of mildew spots on the surface of litchi peel was larger than 10 mm. According to the method by He et al. [23], the litchi pericarp browning degree was defined according to five grades. The determination of anthocyanin content in pericarp was based on the method by Wei et al. [24]. The total soluble solids (TSS; %) content was determined using a handheld digital refractometer (MA150; Dolis, Germany). Titratable acidity (TA; %) was assessed via the titration of juice against sodium hydroxide and expressed as % citric acid. The samples were measured five times and averaged.

### 2.3. Determination of Antifungal Activity of Ag-NP Solution against P. litchi

#### 2.3.1. Antifungal Activity

The antifungal activity of the Ag-NP solution against *P. litchi* was measured using the inhibition zone test based on the plate method. Vernier calipers were used to determine the size of the fungistatic circle, and each group of experiments was performed five times in parallel. We also measured the growth of mycelial radicles, and the activity was expressed as EC50 (effective concentration for 50% of maximum effect) [25]. The EC_50_ value was calculated according to the relationship between the concentration of the Ag-NP solution and mycelial radial growth. First, the percent inhibition of mycelial growth (IMC) was calculated according to the following formula:IMC (%) = (dc − dt)/dc × 100(1)
where dc (mm) is the mean colony diameter for the control sets and dt (mm) is the mean colony diameter for the treatment sets. Second, the inhibition rate was transformed to a probability value (Y), and the concentration of Ag-NP solution (X) was log-transformed. Then, linear regression (Y = *a* + *bx*) was performed, and the coefficient (R^2^) was estimated with interpolation from computer-generated log-probit plots of the concentrations of Ag-NP solutions and relative inhibition. The EC_50_ value was the logarithm value of X when Y = 5.

#### 2.3.2. Observation of *P. litchi* Morphology

The fresh hyphae used for these experiments were collected from 7-day-old cultures of *P. litchi* growing in V8 medium. Hyphae were soaked in Ag-NP solutions at different concentrations (50, 100, 200, 400, and 800 μg/mL) and incubated at 25 °C for 24 h. Then, hyphae were removed and prepared into electron-microscope samples and observed under the microscope (Murzider ZX803; China). Each treatment was repeated five times.

#### 2.3.3. Determination of *P. litchi* DNA Concentration

The fresh hyphae were soaked in Ag-NP solutions at different concentrations (50, 100, 200, 400, and 800 μg/mL) and incubated at 25 °C for 4 h. Hyphae soaked in distilled water were used as the blank control. Total DNA was extracted using TIANamp Bacteria DNA Kit (Code No. DP302; Tiangen, Beijing, China) according to the manufacturer’s guidelines. The DNA concentration was detected using a UV spectrophotometer (Quawell q5000; Foster City, CA, USA) at a wavelength of 260 nm. Each treatment was repeated five times.

#### 2.3.4. Determination of Defense Enzyme Activity

Hyphae (0.1 g) were soaked in 20 mL of 400 μg/mL Ag-NP solution for different time intervals (2 h, 4 h, 6 h, 8 h, and 10 h). The hyphae were rinsed with distilled water for 3 times and dried with adsorbent paper. The fresh hyphae were ground in liquid nitrogen and homogenized with buffers in an ice bath to extract superoxide dismutase (SOD), peroxidase (POD), and catalase (CAT) enzymes. Then, the homogenate was placed in a centrifuge machine at 4 °C and 8000× *g* r/min for 10 min. SOD was detected using a superoxide dismutase (SOD) assay kit (Code No. BC0170; Solarbio, Beijing, China); POD was detected using a peroxidase (POD) assay kit (Code No. BC0090; Solarbio, Beijing, China); and CAT was detected using a catalase (CAT) assay kit (Code No. BC0200; Solarbio, Beijing, China) according to the manufacturer’s guidelines.

### 2.4. Statistical Analysis

A completely randomized design was used in this experiment and repeated in quintuplicate. Data were subjected to analysis using SPSS 19.0 software (Chicago, IL, USA) (*p* < 0.05) and are presented as the mean values and standard deviations.

## 3. Results

### 3.1. Effects of Ag-NP Treatment on the Health Rate, Pericarp Browning Index, and Anthocyanin Content of Litchi

With the extension of storage time, the rate of healthy litchi fruit decreased under the two storage conditions (Figure 1A). It is evident from the results that the Ag-NP solution significantly (*p* ≤ 0.05) decreased decay in litchi stored at 4 °C compared with the control treatments. After fifteen days of storage at 4 °C, the 400 μg/mL Ag-NP-solution-soaked sample lost 35% of its initial health compared with 77% for the samples soaked with clear water. After twenty-five days of storage at 4 °C, the health rate of lychee soaked in 400 μg/mL Ag-NP solution was 20%, and the control groups showed 0%, with significant differences.

With the extension of storage time, the anthocyanin content of the pericarp decreased, and the browning index increased under the two storage conditions (Figure 1B,C). From five to twenty-five days of storage at 4 °C, the anthocyanin content of pericarp soaked in Ag-NP solution at 400 μg/mL decreased by 28.8%. After twenty-five days of storage at 4 °C, the anthocyanin content of pericarp treated with Ag-NP solution at 400 μg/mL was 104% higher than that of clear-water treatment (Figure 1B). After twenty days of storage at 4 °C, the browning index of litchi treated with 400 μg/mL Ag-NPs was 37.6% lower than that of the control (*p* < 0.05) (Figure 1C). Ag-NP treatment had an obvious effect on inhibiting the browning of litchi pericarp.

### 3.2. Effects of Ag-NP Treatment on the Total Soluble Solids (TSS) and Titratable Acidity (TS) of Litchi

We further studied the effects of Ag-NP solution on the quality of litchi under different storage conditions. With the extension of storage time, the contents of TSS and TA in fruits decreased under the two storage conditions (Figure 2A,B). From fifteen to twenty-five days of 4 °C storage, the TSS content of fruits treated with Ag-NP solution was higher than that of the control group, and the effect of 400 μg/mL Ag-NP solution was the best. When stored at 4 °C for twenty-five days, the TSS content of the Ag-NP solution treated with 400 μg/mL was 178% that of clear-water treatment (Figure 2A). The TA content of litchi fruit slowly decreased over the storage period, and there were no significant differences among treatments (Figure 2B).

### 3.3. Effects of Ag-NP Treatment on Mycelial Growth of P. litchi

In plate assays, the Ag-NP solutions had an inhibitory effect on *P. litchi* and inhibited the mycelial growth of *P. litchi* in a dose-dependent manner (Table 1). At 800 μg/mL, the inhibitory effect of the Ag-NP solution was 94.87%, and the Ag-NP solution completely inhibited the mycelial growth of *P. litchi*.

The effective concentration (EC_50_) values of the Ag-NP solutions were estimated using probit analyses. The virulence regression equation was y = 0.0709x + 45.174; the EC50 was 68 μg/mL, and the r^2^ coefficient was 0.8090. The Ag-NP solution had a strong inhibitory effect on *P. litchi*.

### 3.4. Effects of Ag-NP Treatment on Morphology of P. litchi

The effects of different concentrations of Ag-NP solutions on the morphology of *P. litchi* were examined using an optical microscope (Murzider ZX803; Dongguan, China). All sporangia of *P. litchi* treated with 50, 100, 200, 400, and 800 μg/mL Ag-NP solutions for 24 h showed considerable changes in sporangia morphology. The control sporangia grown on sterile water had smooth, slender germ tubes. In contrast, *P. litchi* treated with 100, 200, 400, and 800 μg/mL Ag-NP solution showed deformed germ tubes and sporangial distortion (Figure 3). Sporangial growth was significantly inhibited by the Ag-NP solution when the concentration was 400 μg/mL or more. At this point, the number of spores began to decrease, and the hyphae stopped growing. The inhibitory effect of the Ag-NP solution on the sporangial germination of *P. litchi* was positively related to the concentration of the Ag-NP solution used. Therefore, the highly concentrated Ag-NP solution permeated into spores, which led to spore rupture and loss of infectivity.

### 3.5. Effects of Ag-NP Treatment on the DNA Concentration of P. litchi 

The effects of different concentrations of Ag-NP solutions on the DNA concentration of *P. litchi* were determined using both ultraviolet spectrophotometry and gel electrophoresis. The Ag-NP solution had a significant effect on the content of DNA in *P. litchi* (Figure 4A). The DNA content of *P. litchi* sharply decreased with the increase in the concentration of the Ag-NP solution (Figure 4A). Gel electrophoresis showed that the brightness of the bands with Ag-NP solution was significantly lower than that of the control. The electrophoretic bands almost disappeared with the 800 μg/mL Ag-NP solution (Figure 4B).

### 3.6. Effects of Ag-NP Treatment on SOD, CAT, and POD Activities and Protein Content in P. litchi

SOD, CAT, and POD are three essential defense enzymes used by organisms. They can prevent the production of reactive oxygen and prevent cells from being damaged [26]. Figure 5A shows that the SOD content in *P. litchi* gradually increased and peaked at 6 h, followed by a gradual decrease. The SOD content in Ag-NP-solution-treated *P. litchi* was 6% higher than that in the control at 6 h of soaking (Figure 5A). The CAT and POD contents in *P. litchi* rapidly increased and reached their peaks at 4 h, followed by a sharp decrease in activity. The peak values of CAT and POD in Ag-NP-solution-treated cells were 165.56% and 135.38% higher than those in control cells, respectively (Figure 5B,C). The protein content in *P. litchi* rapidly increased and reached its peak at 4 h, followed by a sharp decrease. The peak protein content in Ag-NP-solution-treated cells was 148.2% higher than that in control cells (Figure 5D).

## 4. Discussion

There are several fields in which Ag-NPs can be used, including antimicrobials, food processing, and packing of fruits and vegetables [27]. Because of their excellent fresh-keeping and antibacterial properties, they are widely used in the field of fruit and vegetable preservation [28,29]. Ag-NPs could reduce the incidence of post-harvest brown rot in apples [27]. Ag-NPs were helpful in maintaining the quality of grapes for at least 30 days and allowed the grapes to live longer [30]. Additionally, the synthesis of Ag-NPs using plants is simple, ecofriendly, and economical [30,31]. According to Kamel [27], Ag-NPs did not express any phytotoxic activity when they were used to treat apple fruit. Thus, Ag-NPs have wide application prospects in bacteriostasis and the fresh-keeping of fruit and vegetables.

Litchi is one of the most difficult fruits to store because it easily browns and rots after harvesting [32]. Determining the anthocyanin content of litchi pericarp is one way to determine the characteristics of litchi freshness. In a previous study, anthocyanin was the main factor affecting the browning of litchi pericarp, which is the main component of the red pigment in litchi pericarp [3]. Thus, the anthocyanin content affects the browning of the fruit to some extent. During storage, litchi treated with 400 μg/mL Ag-NPs and stored at 4 °C had the highest health rate and the lowest browning index of all the samples (Figure 1A,C). These findings are similar to those of the study by Xiao [33] on the preservation of litchi with nanoselenium composites. The decrease in the healthy rate and in the anthocyanin and TSS contents of litchi was due to the accelerated ripening after harvest of the fruit due to the continuous strong respiration of the fruit. Respiration can produce ethylene and consume the nutrients of the fruit. According to the research results obtained by Zhou [34], Ag-NPs can catalyze the oxidation of ethylene, which can reduce the content of ethylene and respiration. In addition to being one of the substrates needed to carry out respiration, the TSS also represent a relevant parameter for determining the flavor of fruits [35]. Accordingly, Ag-NP treatment is effective in reducing the respiration of fruit, in turn reducing the consumption of TSS. To some extent, the original quality of the fruit is maintained.

Meanwhile, Ag-NPs have a high surface-area-to-volume ratio, which can inhibit microorganisms [36]. When bacteria invade the fruit, Ag^+^ ions immediately permeate into the bacterial cells, causing damage to the biofilm and killing the bacteria. Moreover, the Ag-NP solution can form a protective film on the surface of fruit, which acts as a semipermeable barrier against oxygen, carbon dioxide, and moisture, reducing respiration, water loss, and oxidation reactions [37,38,39]. We found that litchi storage at 4 °C with the Ag-NP solution treatment at 400 μg/mL had the best effect and prolonged the shelf life of litchi fruits. A similar result was revealed by Nicolas [40]; the treatment of tomato fruit with Ag-NPs formed a dense protective coat on the surface of the fruits to reduce water vapor transmittance and transpiration and increased the shelf life of the fruits. Devadiga et al. [41] revealed that a chitosan nanosilver coating could prolong the shelf life of apple, tomato, pepper, and eggplant to 25 days, 21 days, 23 days, and 30 days, respectively. As a result, Ag-NPs have the opportunity to be developed into a safe and stable fresh-keeping material.

*P. litchi* infection can cause pericarps decay during litchi fruit preservation. In a previous study, Ag-NPs exhibited broad-spectrum antifungal activity against phytopathogenic fungi. In our study, both the mycelial growth and the sporangial morphology of *P. litchi* were inhibited in the presence of Ag-NP solution, and the inhibitory efficacy was positively correlated with the Ag-NP concentration. The mycelial growth of *P. litchi* was totally suppressed, and the sporangial morphology of *P. litchi* was significantly affected and became abnormal due to the Ag-NP solution. Ag-NPs showed higher antifungal activity against *Bipolaris maydis* at a concentration of Ag-NP solution of 100 μg/mL, and no colony formation was observed on the solid plate [42]. Similarly, Masudulla et al. [43] found that Ag-NPs biosynthesized by plants could inhibit *Ralstonia solanacearum* and *Fusarium oxysporum*. The above studies pointed out that Ag-NPs have significant antifungal activity, which lays the foundation for the comprehensive control of plant pathogens.

The determination of the defense activity test was applied in our study to illustrate the mechanism of antifungal action of Ag-NPs on *P. litchi*. The maintenance of the dynamic balance of defense enzymes plays an important role in clearing up the accumulation of oxygen free radicals in cells. In previous studies, SOD was shown to enter the mitochondrial matrix and prevent the excessive production of reactive oxygen species in cells [44]. CAT and POD can catalyze the excess H_2_O_2_ transformation to H_2_O in cells [45]. When cells are continuously stimulated by the outside world, the balance in the enzymatic system can be destroyed; then, the cells may be damaged. To eliminate reactive oxygen species, the activities of SOD, POD, and CAT in the cell enzymatic system rapidly increased in a short time to increase cell resistance. In our study, SOD, CAT, and POD activities rapidly increased and attained peak values because *P. litchi* was constantly stimulated by Ag-NP solutions, which triggered the defense system and produced many reactive oxygen species. Li et al. [46] found similar results in the treatment of *Fusarium graminearum* with Ag-NPs. After that, the activities of SOD, CAT, and POD began to decline because the continuous accumulation of reactive oxygen species exceeded the cleaning abilities of SOD, POD, and CAT, broke the balance among defense enzymes in cells, and destroyed the integrity of the membrane system [47]. As a result, the Ag-NP solution could disrupt the integrity of the cell membrane and damage the cells of *P. litchi*. At this time, *P. litchi* treated with Ag-NP solutions had a weakened infection ability, which reduced the incidence of litchi downy blight and prolonged fruit shelf life.

## 5. Conclusions

Based on this research study, it can be concluded that Ag-NP solutions could reduce the speed of various reactions of litchi during storage to prolong shelf life, including the health rate, browning index, anthocyanin content, and TSS and TA contents. Additionally, in vitro *P. litchi* mycelial growth and spore germination were heavily inhibited by the Ag-NP solution, and the inhibition rate of the Ag-NP solution at 800 μg/mL against *P. litchi* was as high as 94.87%. Thus, Ag-NPs can potentially be used to control post-harvest litchi downy blight, thus providing viable alternatives to the use of chemical synthetic fungicides. It is hoped that the outcomes of this study generate research on the development of Ag-NP fresh-keeping materials to control the occurrence of post-harvest diseases in fruit and vegetables and prolong their shelf life. As a result, Ag-NPs have the opportunity to be developed into a safe and stable fresh-keeping bacteriostatic agent.

## Figures and Tables

**Figure 1 foods-11-02934-f001:**
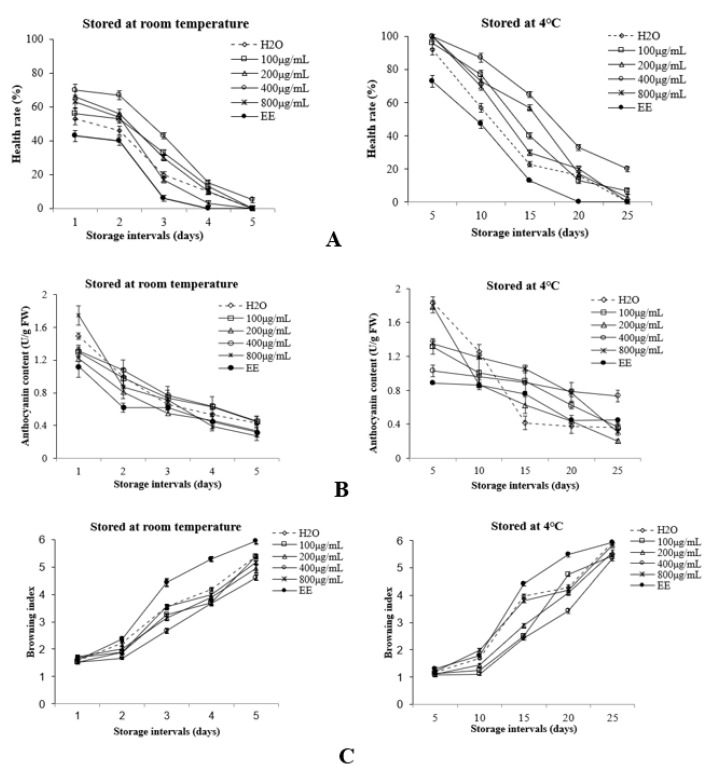
Effects of different treatments on the health rate (**A**), anthocyanin content (**B**), and browning index (**C**) in the pericarp of litchi during storage.

**Figure 2 foods-11-02934-f002:**
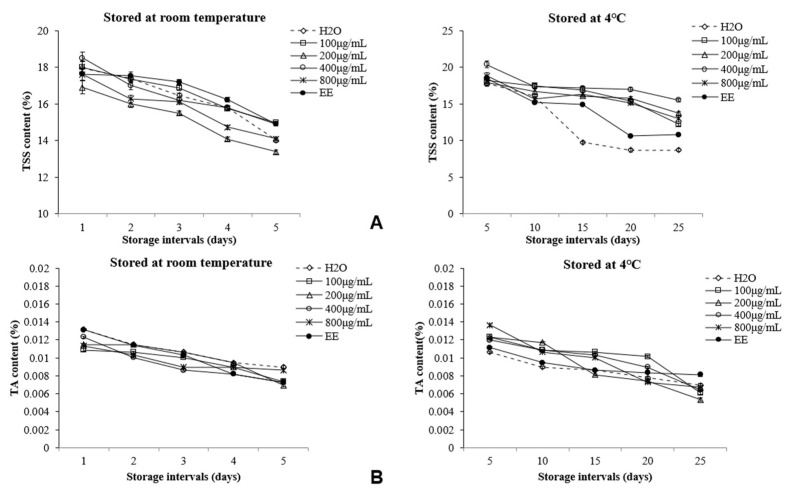
Effects of different treatments on total soluble solids (**A**) and titratable acidity (**B**) contents of litchi during storage.

**Figure 3 foods-11-02934-f003:**
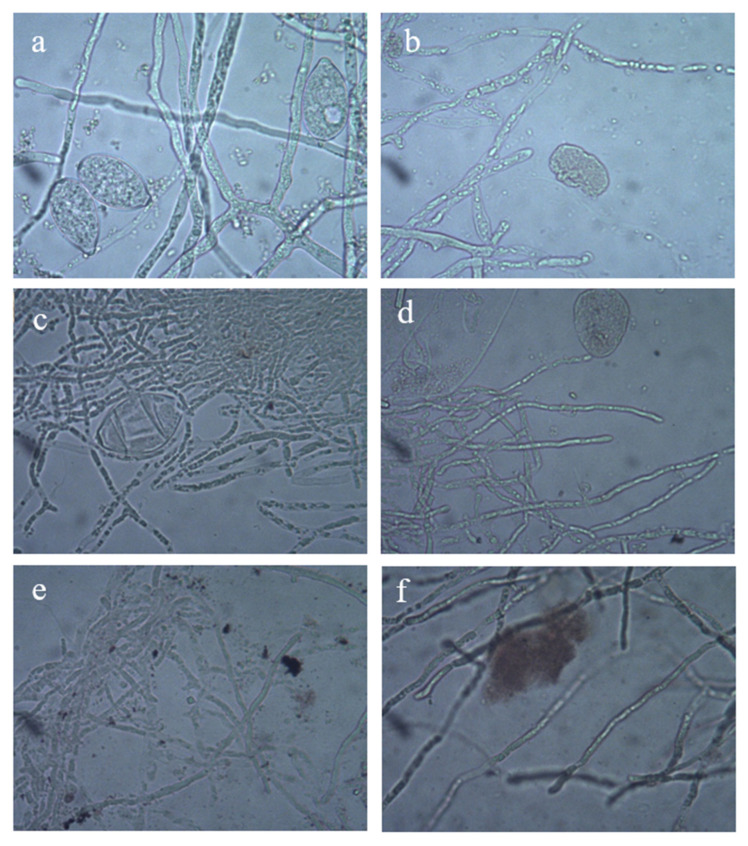
Micrographs of the sporangia of *P. litchi* with or without Ag-NP solution. (**a**) Control; (**b**) 50 μg/mL; (**c**) 100 μg/mL; (**d**) 200 μg/mL; (**e**) 400 μg/mL; and (**f**) 800 μg/mL.

**Figure 4 foods-11-02934-f004:**
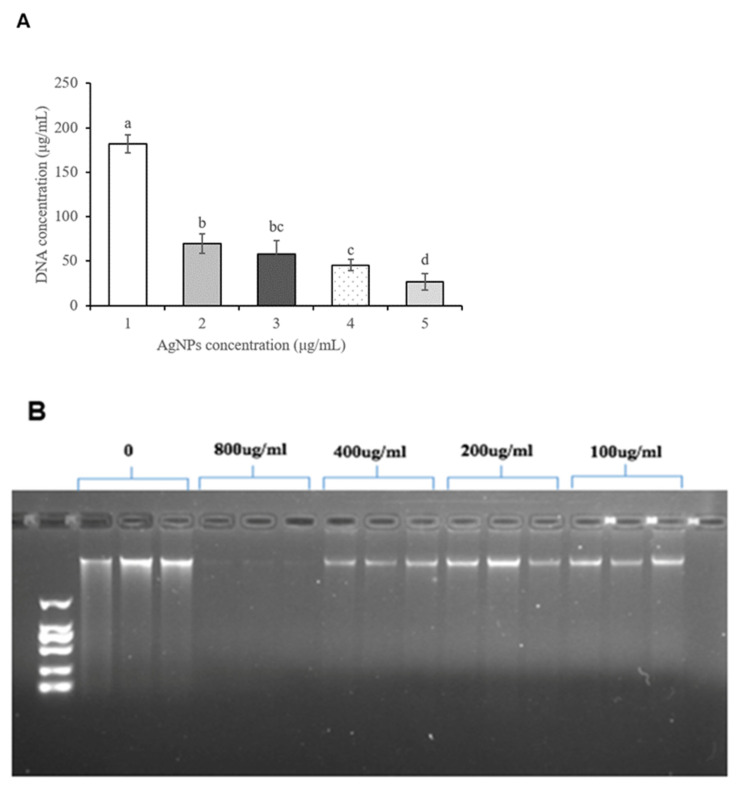
Effects of different treatments on the DNA concentration (**A**) and DNA gel electrophoresis map (**B**) of *P. litchi*. Lower-case letters indicate the significant differences (*p* < 0.05) according to Duncan’s multiple range test.

**Figure 5 foods-11-02934-f005:**
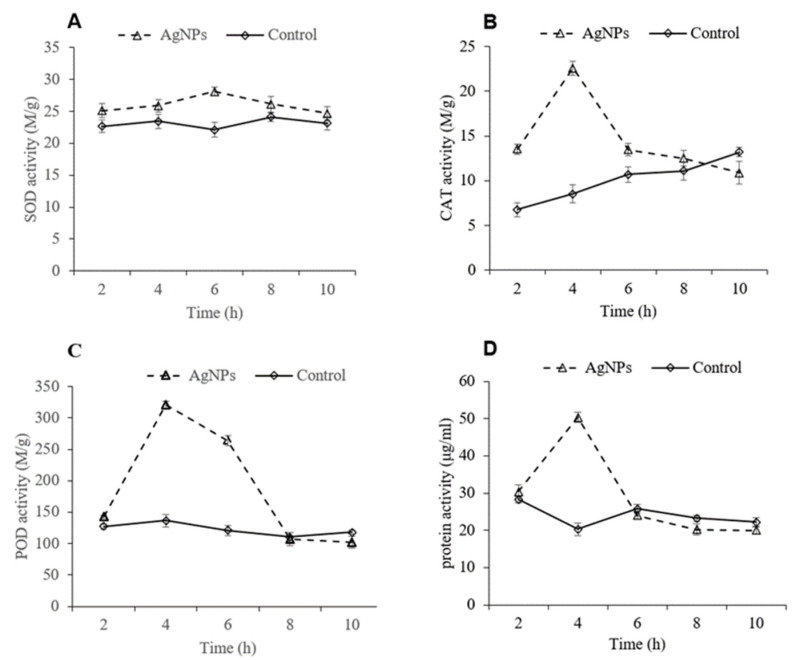
Activities of superoxide dismutase (SOD) (**A**), catalase (CAT) (**B**), and peroxidase (POD) (**C**) and protein concentration (**D**) of *P. litchi* treated with Ag-NPs.

**Table 1 foods-11-02934-t001:** Inhibitory effects of Ag-NPs on *P. litchi* growth.

Concentration of Ag-NPs (μg/mL)	Colony Diameter (mm)	Percentage of Mycelia Growth Inhibition (%)
control	31.81 ± 0.0408	0
25	21.69 ± 0.0510	31.81
50	16.21 ± 0.0510	49.04
100	15.07 ± 0.0510	52.65
200	8.73 ± 0.0510	72.56
400	5.77 ± 0.0510	81.86
800	4.82 ± 0.0510	94.87

## Data Availability

Not applicable.

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
