# Peer review of "Preservation of Litchi Fruit with Nanosilver Composite Particles (Ag-NP) and Resistance against Peronophythora litchi"

_foods, 2022, doi:10.3390/foods11192934_

Round 1

Reviewer 1 Report

Comments and Suggestions for Authors

Authors have done excellent research entitled “Preservation of litchi fruit with nanosiliver (Ag-NP) composite particles and resistance against Peronophythora litchi

Dear authors,

·         The title implies that the manuscript is dealing with the Preservation of litchi fruit with nanosiliver (Ag-NP) composite particles and resistance against Peronophythora litchi. However, the aim of the study is not stated in the abstract and introduction section  (Rows 29-31 – PAGE 1).

·         Make sure that the order of experiments in M&M matches with Results section.

  •  Materials and methods remain incomplete. V8 medium and EC50 value, why did you use 20 mL of 400 μg/mL Ag-NP solution (literature? own experiments?).  
  • Add details of the (SOD, POD and CAT enzymes) (line 169). 
  • Peronophythora litchi' should be mention the first time, ' P. litchi ' the 2nd time during abstract, introduction, ..... sections
  •  The discussion and abstract are just as concise.
  • Remove this section (6. Patents) if there are not patents resulting from 359 the work reported in this manuscript. (L358 -360)

·         Kindly change where into Where (L45) PAGE 4

·         2 g , 5 g, 3 percentage - 4 °C, 2-3 days  .............. Kindly change to two , five , three............

  • Inline 51 page 2, change but to In contrast

·         Spelling mistakes and incomplete sentences are common (absorb, average, where,....

  • Inline 61-64 page 5, the sentence is not very clear (and do not produce drug resistance to pathogenic microorganisms …………..)
  • Inline 312 page 10, the sentence is not very clear (As a 312 result, c)
  • Add details of the (optical microscope) (line 231) page 6

Author Response

Point 1: The title implies that the manuscript is dealing with the Preservation of litchi fruit with nanosiliver (Ag-NP) composite particles and resistance against Peronophythora litchi. However, the aim of the study is not stated in the abstract and introduction section (Rows 29-31 – PAGE 1)

Response 1: Thank you for your suggestion. It has been modified in re-submit manuscript, and the modification was highlighted.

Litchi (Litchi chinensis Sonn.) is susceptible to infection by Peronophythora litchi poststorage, which rapidly decreases the sensory and nutritional quality of the fruit. In this study, the effects of nanosilver (Ag-NP) solution treatment on the shelf life of litchi fruit and the inhibition of P. litchi were examined, and the underlying mechanisms were discussed. For investigations, we used one variety of litchi (‘Feizixiao’), dipping it in different concentrations of Ag-NP solution after harvesting. Meanwhile, we treated P. litchi with different concentrations of Ag-NP solution. According to the data analysis, litchi treated with 400 μg/mL Ag-NPs and stored at 4 °C had the highest health rate and the lowest browning index of all the samples. In the same trend, treatment with 400 μg/mL Ag-NPs produced the best results for anthocyanin content, total soluble solids content and titratable acidity content. Additionally, according to the results of the inhibition test, 800 μg/mL Ag-NP solution had a 94.97% inhibition rate against P. litchi. Within 2-10 h following exposure to 400 μg/mL Ag-NP solution, the contents of Superoxide dismutase, Peroxidase, and Catalase in P. litchi gradually increased and peaked, followed by a gradual decline. At this time, the integrity of the cell membrane of P. litchi could be broken by Ag-NP solution, and the sporangia showed deformed germ tubes and abnormal shapes.  Taken together, these results suggested that Ag-NPs treatment inhibited respiration and P. litchi activity, which might attenuate litchi pericarp browning and prolong the shelf life of litchi. Accordingly, the Ag-NPs could be used as an effective antistaling agent in litchi fruit and an ecofriendly fungicide for postharvest control of litchi downy blight. This study provided new insights of Ag-NPs application in antistaling agent of fruit storage and ecofriendly fungicide.

Point 2: Make sure that the order of experiments in M&M matches with Results section.

Response 2: Thank you for your suggestion. We have revised “2.3 Statistical analysis” to “2.4 Statistical analysis”.

Point 3 Materials and methods remain incomplete. V8 medium and EC50 value, why did you use 20 mL of 400 μg/mL Ag-NP solution (literature? own experiments?)

Response 3: Thank you for pointing out our un-cautious issue in Materials and methods. Firstly, in the experiment of litchi treated with Ag-NP solution, we obtained that the best concentration of Ag-NP solution to keep litchi fresh was 400 μg/mL. For this reason, we pay more attention to the antibacterial activity of 400 μg/mL Ag-NP solution against P. litchi. Secondly, In the early stage, we used 20mLV8 solid medium to culture P. litchi, so we also chose 20mL Ag-NP solution when dealing with hyphae.

Point 4 Add details of the (SOD, POD and CAT enzymes) (line 169).

Response 4: Thank you for your suggestion. We have made correction according to your comments.

Point 5 Peronophythora litchi' should be mention the first time, ' P. litchi ' the 2nd time during abstract, introduction, ..... sections

Response 5: Thank you for your suggestion. We have made correction according to your comments.

Point 6 The discussion and abstract are just as concise.

Response 6: Thank you for your suggestion. We have made correction according to your comments (Line311-313).

Point 7 Remove this section (6. Patents) if there are not patents resulting from 359 the work reported in this manuscript. (L358 -360)

Response 7: Thank you for your suggestion. We have removed this section according to your suggestion.

Point 8 Kindly change where into Where (L145) PAGE 4

Response 8: Thank you for your suggestion. We have made correction according to your comments.

Point 9 2 g , 5 g, 3 percentage - 4 °C, 2-3 days .............. Kindly change to two , five , three...........

Response 9: Thank you for your suggestion. We have made correction according to your comments.

Point 10 Inline 51 page 2, change but to In contrast

Response 10: Thank you for your suggestion. We have made correction according to your comments.

Point 11 Spelling mistakes and incomplete sentences are common (absorb, average, where,..

Response 11: Thank you for your suggestion. We have made correction according to your comments.

Point 12 Inline 61-64 page 5, the sentence is not very clear (and do not produce drug resistance to pathogenic microorganisms …………)

Response 12: Thank you for your suggestion. It is really true as you suggested that this statement may unclear to the readers. We have revised this sentence to: “Additionally, Ag-NPs have strong inhibition and killing effects on dozens of pathogenic microorganisms [15] and do not produce drug resistance”.

Point 13 Inline 312 page 10, the sentence is not very clear (As a 312 result, c)

Response 13: Thank you for your suggestion. It is really true as you suggested that this statement may unclear to the readers. We have supplemented this sentence to: “As a result, Ag-NPs have the opportunity to be developed into a safe and stable fresh-keeping material”.

Point 14 Add details of the (optical microscope) (line 231) page 6

Response 14: Thank you for your suggestion. We have made correction according to your comments (line 250).

Reviewer 2 Report

Lin et al. have synthesized AgNPs by using Phyllanthus emblica and evaluated their inhibition of Peronophythora litchi for preserving litchi fruit and extending the shelf life. This is an interesting piece work carried out systematically and well-written. Therefore, I have the following minor comments:  

1.           One-time used “total soluble solids” and “titratable acidity” within abstract need not be abbreviated as TSS and TA respectively.

2.           SOD, POD and CAT should be provided in the full form and there is not need for abbreviation in the abstract.

3.           “mechanisms that underlie” should be rewritten as “underlying mechanisms” in the introduction.

4.           “Phyllanthus emblica” should be italicized throughout the manuscript.

5.           Why a reference citation not provided for section 2.1.1. as authors indicated that they reported in their previous study.

6.           Why same sub-title for sections 2.1.2 and 2.1.3?

7.           How did the authors get the crude polysaccharide aqueous solution from Phyllanthus emblica?

8.           How was the concentration of AgNPs determined to prepare different concentrations ranging from 25-800 mg/mL?

9.           “Vernier caliper” is a two word and not a single word.

10.        All the methods given in section 2 should include a reference citation each.

11.        “State, city and country” details should be provided for SPSS 19.0 software in section 2.3 if it is from USA, while just “city and country” for other countries. This rule can also be followed for other materials/reagents/chemicals/instruments details in the section 2.

12.        Figures 1-5 – all the abbreviations used in these figures should be provided in full form in the respective figure captions.

13.        Wherever applicable, different treatments should be detailed in under the respective figure caption.

14.        Conclusion – a final line highlighting the future extension of this work should be included.

15.        At least 8-10 old and irrelevant references should be removed.

Author Response

On behalf of my co-authors, we thank you very much for giving us an opportunity to revise our manuscript, we appreciate you very much for the positive and constructive comments and suggestions on our manuscript.

Response to Reviewer 2 Comments

Point 1 One-time used “total soluble solids” and “titratable acidity” within abstract need not be abbreviated as TSS and TA respectively.

Response 1: Thank you for your suggestion. We have made correction according to your comments.

Point 2 SOD, POD and CAT should be provided in the full form and there is not need for abbreviation in the abstract.

Response 2: Thank you for your suggestion. We have made correction according to your comments.

Point 3 “mechanisms that underlie” should be rewritten as “underlying mechanisms” in the introduction.

Response 3: Thank you for your suggestion. We have revised “mechanisms that underlie” to “underlying mechanisms”.

Point 4 “Phyllanthus emblica” should be italicized throughout the manuscript.

Response 4: Thank you for pointing out this problem. We have made correction according to your comments, and also check rest of the paper.

Point 5 Why a reference citation not provided for section 2.1.1. as authors indicated that they reported in their previous study.

Response 5: Thank you for your suggestion. We have added the references in this section.

Point 6 Why same sub-title for sections 2.1.2 and 2.1.3?

Response 6: Thank you for your suggestion. 2.1.2 sub-title is Ag-NP treatment of litchi and 2.1.3 sub-title is Ag-NP treatment of P. litchi. There are different.

Point 7 How did the authors get the crude polysaccharide aqueous solution from Phyllanthus emblica?

Response 7: Thank you for pointing out our negligence in this part. We have supplemented the preparation process of crude polysaccharides aqueous solution in 2.1.1 Prearation of Ag-NP solution.

According to the method of Liao et al. [20], the fresh and disease-free fruit of Phyllanthus emblica were washed, dried, enucleated and freeze-dried, and the powder was obtained after sifting through 40 meshes. The supernatant obtained from the powder extracted by hot water for five hours was dehydrated step by step by ethanol and dried repeatedly with ether to obtain the crude polysaccharide.

Point 8 How was the concentration of AgNPs determined to prepare different concentrations ranging from 25-800 mg/mL?

Response 8: Thank you for pointing out our negligence in this part. We have corrected and supplemented in 2.1.3. Ag-NP treatment of P. litchii.

Ag-NPs was prepared into solutions of 0.5, 1, 2, 4, 8 and 16 mg/mL with distilled water. In the bacteriosttic test, a total of 120 mL of V8 culture medium was poured into 6 culture plates, and 1 mL was taken out from the medium, and the different concentraions (0.5, 1, 2, 4, 8 and 16 mg/mL) of Ag-NP soultion by 1mL were add. Then the final Ag-NP soultion of V8 culture medium was 25, 50, 100, 200, 400 and 800 ug/mL.

Point 9 “Vernier caliper” is a two word and not a single word.

Response 9: Thank you for pointing out this problem. We have made correction according to your comments.

Point 10 All the methods given in section 2 should include a reference citation each.

Response 10: Thank you for pointing out this problem. We have supplemented according to your comments.

Point 11 “State, city and country” details should be provided for SPSS 19.0 software in section 2.3 if it is from USA, while just “city and country” for other countries. This rule can also be followed for other materials/reagents/chemicals/instruments details in the section 2.

Response 11: Thank you for pointing out this problem. We have made correction according to your comments, and also check rest of the paper (line194).

Point 12 Figures 1-5 – all the abbreviations used in these figures should be provided in full form in the respective figure captions.

Response 12: Thank you for pointing out this problem. We have made correction according to your comments.

Point 13 Wherever applicable, different treatments should be detailed in under the respective figure caption.

Response 13: Thank you for pointing out this problem. The caption of figure 3 revised to “Micrographs of the sporangia of P. llitchi. with or without AgNP solution. (a) Control; (b) 50 μg/mL; (c) 100 μg/mL; (d) 200 μg/mLmg/L; (e) 400 μg/mL; and (f) 800 μg/mL”.

Point 14 Conclusion – a final line highlighting the future extension of this work should be included.

Response 14: Thank you for pointing out this problem. We have supplemented according to your comments.  

We added “As a result, Ag-NPs have the opportunity to be developed into a safe and stable Fresh-keeping bacteriostatic agent” at the end of Conclusion.

Point 15 At least 8-10 old and irrelevant references should be removed.

Thank you for pointing out this problem. We have removed the old and irrelevant references. The references of “[2], [6], [7], [26], [29], [39], [44] and [54]” have been removed.